# Hot Exoplanetary Atmospheres in 3D

**William Pluriel** 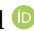

Département d'Astronomie Chemin Pegasi 51, Observatoire Astronomique de l'Université de Genève, CH-1290 Versoix, Switzerland; william.pluriel@unige.ch

**Abstract:** Hot giant exoplanets are very exotic objects with no equivalent in the Solar System that allow us to study the behavior of atmospheres under extreme conditions. Their thermal and chemical day–night dichotomies associated with extreme wind dynamics make them intrinsically 3D objects. Thus, the common 1D assumption, relevant to study colder atmospheres, reaches its limits in order to be able to explain hot and ultra-hot atmospheres and their evolution in a consistent way. In this review, we highlight the importance of these 3D considerations and how they impact transit, eclipse and phase curve observations. We also analyze how the models must adapt in order to remain self-consistent, consistent with the observations and sufficiently accurate to avoid bias or errors. We particularly insist on the synergy between models and observations in order to be able to carry out atmospheric characterizations with data from the new generation of instruments that are currently in operation or will be in the near future.

**Keywords:** planets; exoplanets; hot Jupiters; atmospheres; radiative transfer; atmospheric dynamics; spectroscopy

## 1. Introduction

Giant planets are diverse and complex objects [1–4]. Thanks to the hundreds of hot giant exoplanets discovered so far, the study of their atmospheres is at the forefront of exoplanet research. Spectroscopic observations are now being used to probe these worlds in the search of the molecular features, physical properties and dynamics of their atmospheres. Thanks to recent space (JWST) or ground-based (Espresso, NIRPS, CRIRES) instruments, a sort of revolution is occurring in the field since we will observe more planets in one year than has ever been observed by Hubble and Spitzer! Moreover, we are at the edge of measuring accurate abundances and accurate radial velocity and are working with temporal resolutions high enough to see the impact of the 3D structures of these exoplanets' atmospheres. Such studies are crucial in the pursuit of understanding the diverse nature of the chemical compositions, atmospheric processes and internal structures of exoplanets, as well as the conditions required for planetary formation.

In recent years, there has been a surge in transit spectroscopy observations using both space-borne and ground-based facilities, resulting in significant advancements in our understanding of exoplanetary atmospheres. Transit spectroscopy has been used for the detection of multiple molecular absorption features, including water, methane, iron and, more recently, carbon dioxide [5,6]. Many instruments have been used to pursue such atmospheric characterizations. We can mention in particular the Hubble Space Telescope (HST) in the realm of low-resolution spectroscopy. This telescope was widely used to study hot and ultra-hot Jupiters having enough planets to start population studies and global characterization [7,8]. Many exoplanets have also been studied in the emission and in phase curves, leading to important advances in the characterization of their atmospheres (cloud cover, thermal structure and detection of species) such as the first 3D analyses of a hot Jupiter using phase curves of the highly irradiated planet Wasp-43 b [9]. In most of these studies, uniform atmospheres are assumed, e.g., a column a atmosphere is generalize to

represent the whole planet. Thanks to analyses of both emission and transmission spectra, we obtain information on the two faces of such atmospheres.

However, we know from the Solar System that giant planets present strong longitudinal, latitudinal and temporal variations, highlighting the importance of taking into account the 3D-structure of exoplanetary atmospheres. As an example, the JUNO mission revealed an unexpected complexity of the poles of Jupiter [10], which is like looking at two completely different planets when looking at Jupiter from its pole or from its equator. As it is shown in Figure 1, the polar view of Jupiter shows eddies, depression and anticyclone, whereas the equatorial view reveals large east–west jet streams with way less turbulence. This example from our Solar System highlights the strong latitudinal and longitudinal variation in planetary atmospheres. We could also show the vertical variations (in temperature, composition, cloud deck, etc.) which add even more complexity in such giant atmospheres. This complexity of giant planets is emphasized when we look at hot giant exoplanets by their particular orbital and radiative conditions. Many recent studies [11–18] thus add a caveat on the current uniform assumptions made to study hot and ultra hot Jupiter atmospheres explaining the limits of such assumptions and trying to understand their impact on the observables (phase curve, transmission and emission spectra).

Atmospheric characterization is also carried out using high-resolution spectroscopy instruments [19–21]. With these instruments, we can discretize radial velocity during transit, thus obtaining measurements of the wind speed in the atmosphere [22]. Such a constraint is essential to improve our understanding of exo-atmospheric dynamics, as we will see in this review. Thanks to the current large telescopes, we have achieved sufficient precision to be able to resolve the transit in time. This avoids integrating the spectrum over the entire transit and allows us to obtain constraints during the transit. This temporal resolution is, however, limited by the signal-to-noise ratio.

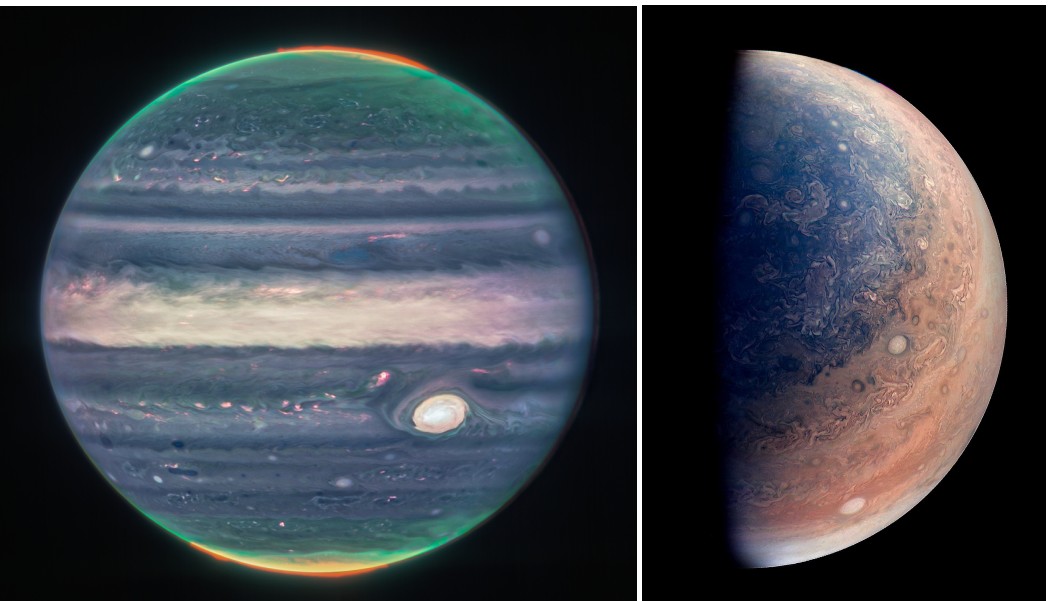

**Figure 1.** (**Left**) Jupiter's atmosphere at the equator observed with JWST/NIRCam. (**Right**) South pole of Jupiter observed by JUNO/NASA. We see the differences in the dynamics between the poles and the region at lower latitudes. When large east–west jet stream structured the planet from the equator to the tropics, the poles show a lot of turbulence and storms without global structures. This is due to the high variation in the atmosphere (in term of dynamics, composition, clouds, etc.) according to latitude and longitude, which thus cannot be reproduced well using 1D models.

In this review, we will focus on hot and ultra-hot Jupiters, showing what theoretical insight the 3D nature of such irradiated atmospheres might bring and what it tells us about their atmospheres. We will see in Section 2 what the current state-of-the-art is in giant exoplanet observations and what the next generation of space- and ground-based instruments will bring to the field. Then, in Section 3, we will develop the interest of the 3D assumption and its observational constraints, showing why the 1D atmospheric assumption was first considered, what its relevance is and what its limits are. We will also explore the different methods and techniques which allow us to obtain information all around the atmosphere. In Section 4, we will then identify how the 3D information obtained allows us to update the recent global climate models (GCMs). We will see in Section 5 how the 3D nature of an atmosphere impacts the interpretation of the data, in particular the retrievals analysis. In Section 6, we will finally discuss the future of this field and the synergy between giant exo-atmospheres and giant planets in the Solar System.

## 2. Giant Exoplanet Observations

The first exoplanet discovered around a main-sequence star was a hot Jupiter orbiting very close to its host star 51 Peg (0.0527 au, [23,24]), a G-type star. Many other hot Jupiters have been detected since then. As these planets orbit close to their star, the probability that they transit in front of them is high :

$$p = \frac{R_*}{a_p(1 - e^2)},$$

(1)

where $p$ is the transit probability, $R_*$ is the stellar radius, $e$ is the planet's orbital eccentricity and $a_p$ is its semi-major axis [25,26]. These planets are thus very interesting targets for transmission spectroscopy. To characterize their atmospheres, the idea is to analyze the light coming from the star filtered by the atmosphere during the primary transit, as well as the light emitted by the atmosphere during the secondary transit. Figure 2 summarizes the phase curve of a transiting planet showing the primary transit and the secondary eclipse. We denoted in red and black which phases correspond to the day and the night sides of the planet, respectively, to highlight which part of the atmosphere is being observed during the phase curve. We will discuss this in more details in Section 3.

Analyzing the phase curve and the primary and secondary transits allows us to determine the composition (species, metallicity, cloud coverage), the physical properties (pressure–temperature profile, emitted flux, albedo) and also the dynamics (wind speed, jet) of the atmosphere of the planet depending on the used instruments. Since the first discovery of exoplanets, we have detected many different species (molecules, atoms and ions). We regroup the main detected species in Figure 3, adapted from Guillot et al. [27]. We see in this figure that we have detected water in many exoplanets. This species should indeed be present in many atmospheres according to equilibrium chemistry [28]. We also note an instrumental bias because the majority of the detection was performed by HST/WFC3, the wavelength of which is centered on a water absorption band (0.8 to 1.7 microns). Few other molecules and atoms were also detected, such as helium, carbon monoxide, sodium, potassium, hydrogen, iron, magnesium and calcium, thanks to low-resolution spectroscopy [5,8,29–32] and HRS [21,22,33–35]. Currently, there is a low significance to most molecular detection data due to the low-resolution power of the instruments and their low sensitivity. Indeed, there is not enough time available and not enough instruments to observe each target several times; thus, it is hard to check each potential detection. That is why only few targets have a high confidence level of detection, as shown in Figure 3.

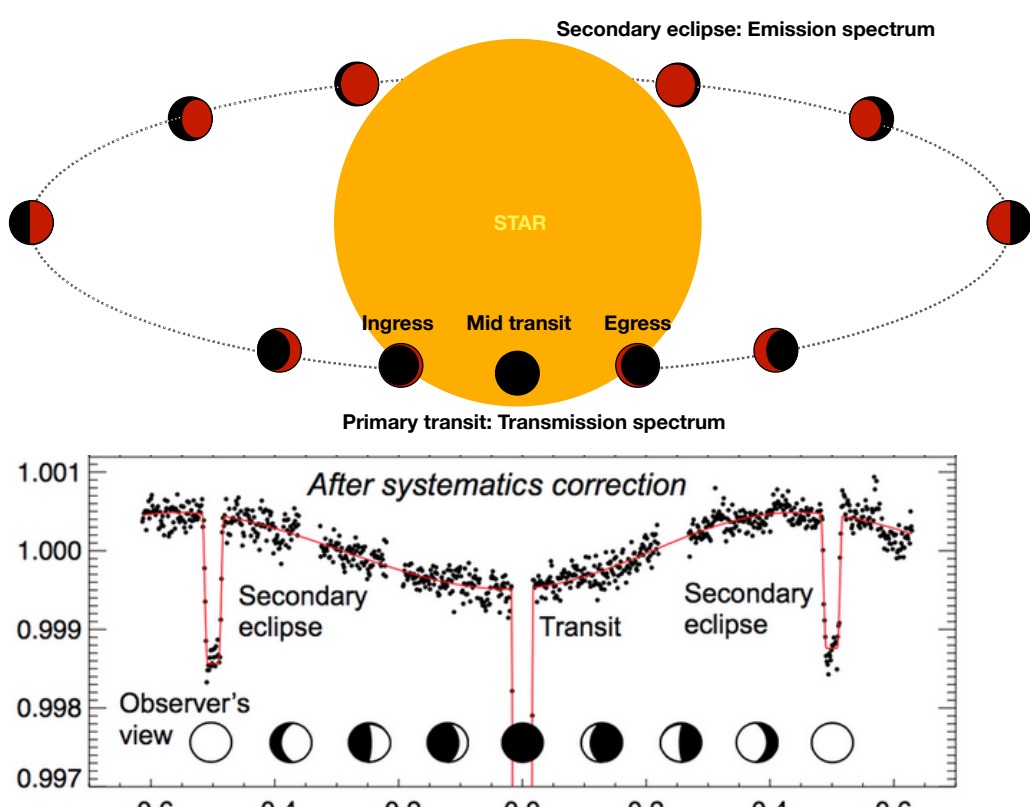

**Figure 2.** (**Top**) An example of the transiting phase curve of the hot Jupiter HD189733 b observed at 4.5 micron with IRAC/Spitzer. During the primary transit, the limbs of the atmosphere are probed just before the secondary eclipse on the day side of the planet is seen. (**Bottom**) Scheme of a transiting exoplanet around its star. The color bar of the planet highlights the day side of the planet (red) and its night side (black). Figure adapted from [36,37].

However, we have recently seen significant improvement in this last area thanks to the introduction of many instruments conducting space- or ground-based observations in the visible or infrared spectra. We regroup in Figure 4 the main telescopes/instruments (low- or high-resolution spectroscopy) capable of performing atmospheric characterization in the visible and in the infrared spectra. Since 2010, we have been listing many instruments for atmospheric characterization, with strong growth since 2016 (12 instruments), including four major instruments in 2022: the high-resolution spectrograph NIRPS at the 3.6 m telescope (ESO) and the low-resolution MIRI, NIRSpec and NIRISS in the JWST (NASA/ESA). We thus expect major improvements in the coming months and years thanks to the high accuracy of the new-generation of instruments able to perform atmospheric characterization. We expect to confirm many species already detected as well as discover many other molecules in exoplanetary atmospheres.

Recently, observations of the ultra-hot Jupiter WASP-76b from CARMENES [38] and ESPRESSO [39] have spatially resolved the features of $H_2O$, HCN and Fe during the planet's transit, showing significant asymmetric chemical distribution in the atmosphere [21,40]. These new-generation high-resolution spectrogaphs indeed allow us to obtain the spatial characterization of atmosphere. In low-resolution spectroscopy, we also report the detection of $CO_2$ (confidence level at 26 sigma) in the hot Jupiter WASP-39b by JWST/NIRSpec [6], which may also be confirmed by other instruments. Indeed, most recent instruments are focused on infrared observation, which is the the most efficient way to conduct atmospheric characterization. In particular, instruments on the JWST allow for a very large wavelength coverage, from 0.6 to 28 microns (with about a 100 spectral resolution), which will be the first instrument to perform an observation in such a large wavelength range at this level of resolution and accuracy. This would help us to obtain accurate elementary abundances

and accurate C/O ratios and metallicities or to better understand the cloud compositions of these exoplanets [41–43].

**Figure 3.** Chemical species detected in exoplanet atmospheres as of October 2022. Confidence level of the detection is shown in color. Few species have been strongly detected so far. The bottom right corner of this table is empty, which may be linked to condensation of the species. Figure adapted from Guillot et al. [27], where the citations for the detection are regrouped in its appendix. We added $CO_2$ detection on WASP-39b [6] and H detection on GJ436b [44].

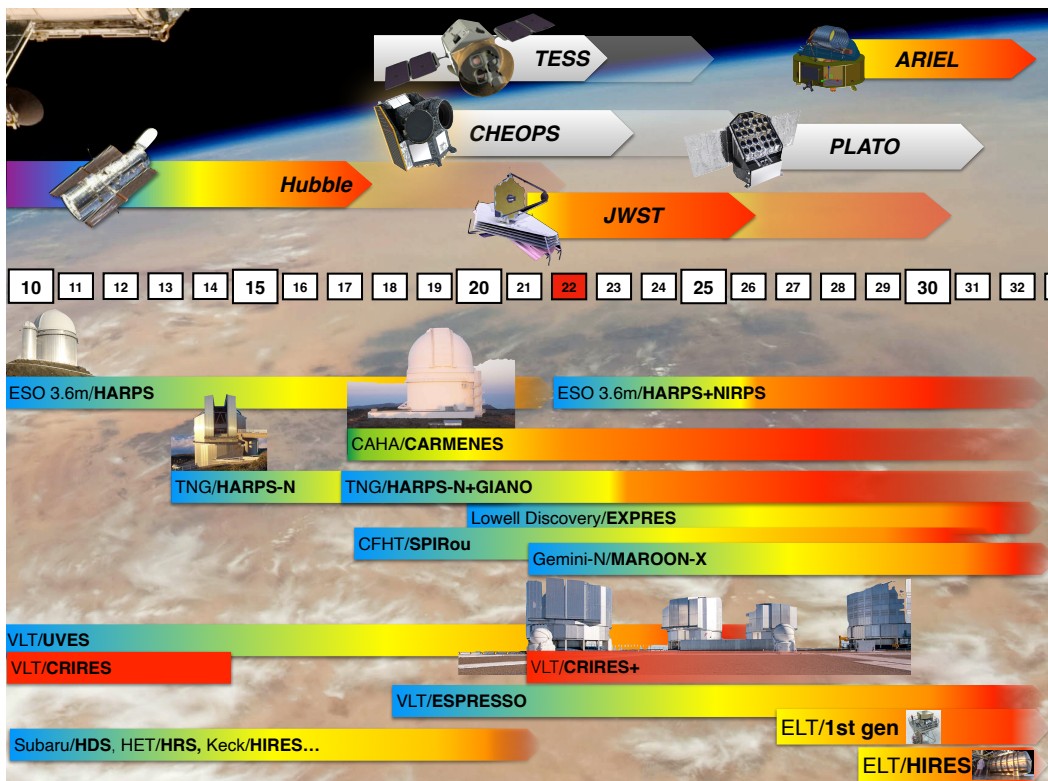

**Figure 4.** Space- and ground-based telescopes/instruments able to perform atmospheric characterization since 2010. We also show the future telescopes/instruments that will come into use during this decade. We see that the number of telescopes/instruments is becoming very large, in particular since 2016, with many overlapping that can simultaneously observe targets with space- or ground-based instruments. Courtesy of David Ehrenreich.

We show in Figure 5 the spectra of several hot Jupiters with the data from HST and Spitzer [7] available as of 2016 compared to a simulated JWST/NIRSpec observations of these WASP-17b atmosphere's using Pandexo [45]. We see that the wavelength range of the NIRSpec covers almost the entire range covered by the other instruments (STIS, WFC3 and IRAC) from 0.6 to 5 microns, with a much better resolution, especially compared to the two Spitzer points. In addition, a whole part of the spectrum from 1.7 to 3.5 microns can be observed, whereas no previous observations covered this wavelength range. It is clear that the resolution combined with the wavelength coverage will bring many more constraints to characterizing these atmospheres. The combination of several instruments could indeed bring more uncertainty or errors than improvements in the atmospheric characterization [46], which is why JWST instruments with broad wavelength coverage are needed. For instance, an analysis of the ultra-hot Jupiter Kelt-7b with HST/WFC3 alone and a combination of HST and Spitzer data showed that the two Spitzer points do not bring more constraints to the retrieval analysis in terms of the molecular detection, the abundance or the T-P profile [47]. This suggest that we need to remain cautious in the combination of data. A very recent NIRSpec observation managed indeed to clearly detect $CO_2$ in the mid-infrared range in the hot Jupiter WASP-39b [6], where the two Spitzer points (3.6 and 4.5 microns) did not have a sufficient resolution to break the degeneracy between $CO_2$ and CO bands [48].

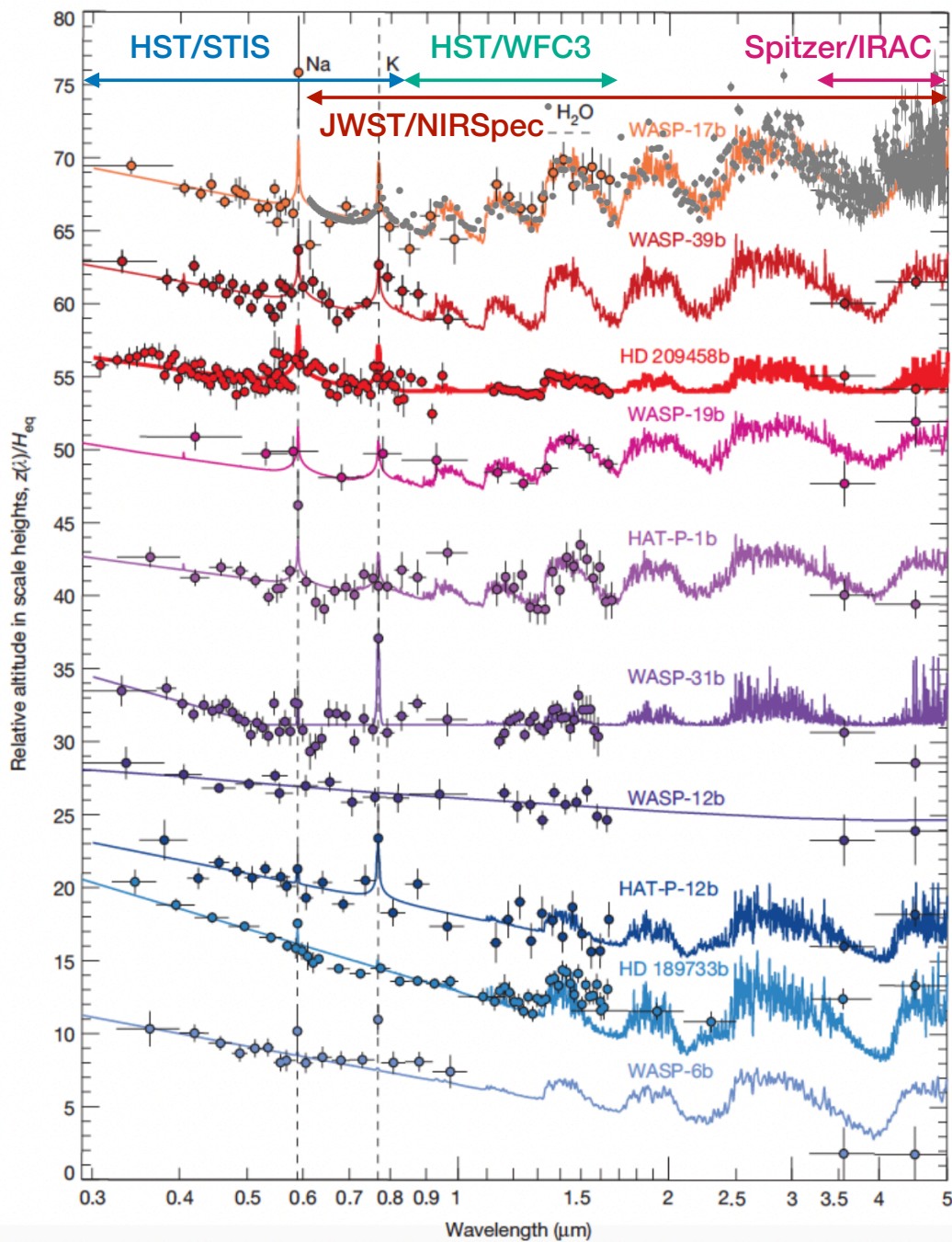

**Figure 5.** HST/Spitzer transmission spectral sequence of hot Jupiter survey targets from Sing et al. [7]. We sub-plotted in gray a JWST/NIRSpec simulation from Pandexo [45] (50 ppm minimum noise, prsim mode with a resolution of 100) on the first spectrum (WASP-17b) to highlight the major improvements in terms of resolution and wavelength coverage. Solid colored lines indicate the fitted spectra from simple 1D models showing the main spectral features. The spectra have been shifted for a better visualization. Horizontal and vertical error bars indicate the wavelength spectral bin and $1\sigma$ measurement uncertainties, respectively. We have also added the wavelength coverage of the different instruments (STIS, WFC3, IRAC and NIRSpec) to emphasise that NIRSpec alone covers the wavelengths of almost all the other instruments combined. Furthermore, NIRSpec has a much better resolution than the other instruments (in particular, the two points of Spitzer/IRAC) and covers a broad band between 1.7 and 3.5 microns that was not observed by the other instruments. Figure adapted from Sing et al. [7].

## 3. From Uniform Constraint Assumption to 3D Atmospheres

Exoplanets are three-dimensional, and we now have instruments that are precise enough to achieve a spatial resolution that forces us to stop using too simplistic assumptions in order to obtain consistent characterizations of their atmospheres. Many improvements have also be made in the theoretical works and simulations that take into account these 3D aspects. In this section, we will focus on the observational improvements.

### 3.1. Phase Curve

The observation of the phase curve of an exoplanet (i.e., the time-dependent change in the brightness of a planet as seen from Earth during one orbital period) is the most straightforward way to probe the planet's longitudinal structure. The brightness of the planet is determined by the combined emitted and reflected light in the observational wavelength. Moreover, we can also observe non-transiting planets with these observations, which significantly enlarges the number of planets that can be observed [49]. Figure 6 shows the phase curve of HD209458b observed by Spitzer/IRAC at 4.5 microns. The amplitude of the phase indicates if an atmosphere is present, and the phase curve offset indicates the brightness distribution. For instance, a telluric planet without an atmosphere will have a typical phase curve with a very high amplitude and no offset since there is no way for the energy around the planet to be redistributed. Furthermore, the difference between the stellar flux (i.e., during the secondary eclipse) and the received flux assuming no primary transit yields the emitted light from the night side, which can be linked to the night side's temperature.

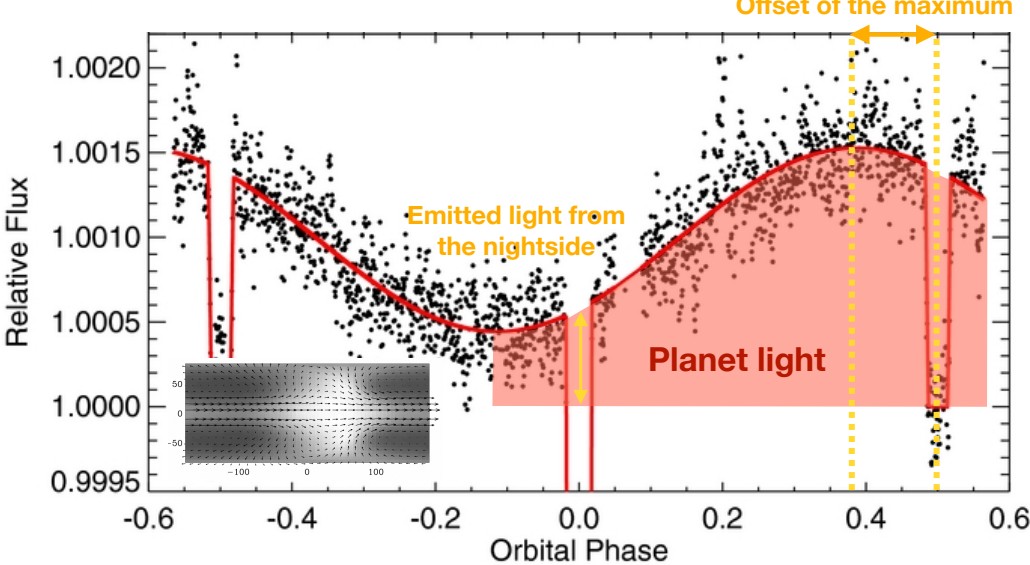

**Figure 6.** Hot Jupiter HD209458b's phase curve observed by Spitzer at 4.5 μm. We show the maximum offset of the phase curve, which is linked to the hot spot offset of the atmosphere. We can deduce the emitted light from the night side looking at the difference between the phase curve if there is no primary transit and the received flux from the star only (i.e., during the secondary eclipse). (**Bottom left**) The longitudinal temperature distribution with the winds (black arrow) from a GCM simulation showing a large equatorial jet and a ~30° hot spot shift [50]. Figure adapted from [51].

Phase curve observations are inherently 3D and are therefore complex to analyze, as several interpretations can explain the data. This is, for example, the case for the ultra-hot Jupiter WASP-12b, which was observed by Spitzer/IRAC at 4.5 microns and also by HST/WFC3. The first IRAC analysis gives two scenarios compatible with the phase curve observations: (i) a solar composition and short-scale height at the terminator or (ii) a solar composition and modest temperature inversion [52]. With the STIS data, the observations fit to either $H_2O$ or $CH_4$ and HCN for an O-rich or C-rich atmosphere, respectively [9].

A re-analysis of these data using GCMs found that the extracted spectrum is steeper than expected, without a clear explanation for this [53].

In the case of hot Jupiters, the impact of clouds is very important in phase curve analyses. Depending on the temperature of the day side and the composition of the clouds ($CaTiO_3$, $MgTiO_3$, MnS and $Na_2S$), the appearance of the planetary day side in the phase curve seems to follow the same trend: clouds cover the whole day side at low equilibrium temperatures then disappear from the eastern part of the day side when the temperature becomes too hot and are finally pushed toward the western limb, where they remain even at greater equilibrium temperatures [54]. The equilibrium temperature at which the transition from a fully cloudy to a partially cloudy planet occurs is a function of the condensation temperature of each species. It has also been demonstrated that the phase curve amplitude does not directly constrain the day to night TP profiles due to the cloud coverage and dynamics [27,54]. Furthermore, the phase curve offset (see Figure 6) does not necessarily track the planetary hot spot offset, particularly when clouds are present on the night side. They suggest that secondary eclipse mapping could be a more robust way to determine the longitude of the hottest point on the planet. We can thus say that the phase curve reveals how the energy is redistributed around the planet.

Currently, full phase curves remain the best method to obtain longitudinally resolved absolute emission spectra of exoplanets. This method, however, remains limited to the transit and the eclipse spectrum analysis by the signal-to-noise ratio of the instruments, their spectral resolution, their wavelength range and also by the models we use to interpret the data (GCMs, retrieval, etc.). The phase curve puts constraints on the flux variability of hot Jupiters, which is precious information that can be used to improve 3D GCM models. We will describe this methodology in Section 4.

### 3.2. Transiting Planets

As they orbit close to their star, a relatively large fraction (∼1%) of hot Jupiters are transiting in front of their host star, which allows us to perform transit spectroscopy. As described in Section 1, we can observe exoplanetary atmospheres during the primary transit (transmission) and during the secondary transit (emission), which give information on the limb and on the day side of the atmosphere, respectively.

In emission, we retrieve a lot of valuable information. First, we can deduce the temperature–pressure profile of the day side of the atmosphere. In particular, we can detect thermal inversion in the atmosphere as it has been claimed for several planets both in low- and high-resolution spectroscopy [55–59]. Such detection remains complicated to confirm, and many have been the subject of debate [51,60–62]. Secondly, we can recognize molecular features from species which allow us to claim detection. Indeed, each species has specific emission lines and bands, combined with a non-isothermal TP-profile which makes the emission spectrum from the day side differ from a black body emission. The detection of these are, however, limited, as explained in Section 2 (see also Figure 3), by the low number of observations and instruments and by the wavelength coverage and the accuracy of the instruments. Finally, emission spectroscopy targets the deeper layer of the atmosphere around a pressure of 0.1 to 1 bar, where the majority of the thermal lines originates. A limit for emission spectroscopy analyses concerns the models used to perform the interpretation, which mostly make 1D assumptions, e.g., they consider that one column of the atmosphere represents the whole atmosphere well (or half-atmosphere in this particular case of emission). This assumption is valid since the resolutions of the instruments do not spatially resolve the day side; thus, the data we obtain come from an integrated flux from the entire day side. Nevertheless, it has been shown that this assumption reaches its limits, especially for hot Jupiters [11,63–65].

As we explained in Section 3.1, the cloud coverage can also be accessible in emission and actually has a great impact on the shape of the emission spectrum [27,54]. Indeed, the cloud coverage depends on the equilibrium temperature and on the species that condensate, the composition of which modifies the emitted flux by several order of magnitudes.

In transmission, we analyze the light coming from the star through the atmosphere of the exoplanet, which provides information on the atmospheric properties at the limb, which is often intuitively assumed to be a narrow annulus around the planet. Indeed, for planets with a $T_{eq} < 1000$ K, the atmospheres are supposed to efficiently redistribute the irradiation received on the day side thanks to atmospheric circulation resulting from a homogeneous atmosphere [3,66,67]. In this case, it is reasonable to assume that a column of the atmosphere at the limb represents the whole atmosphere well.

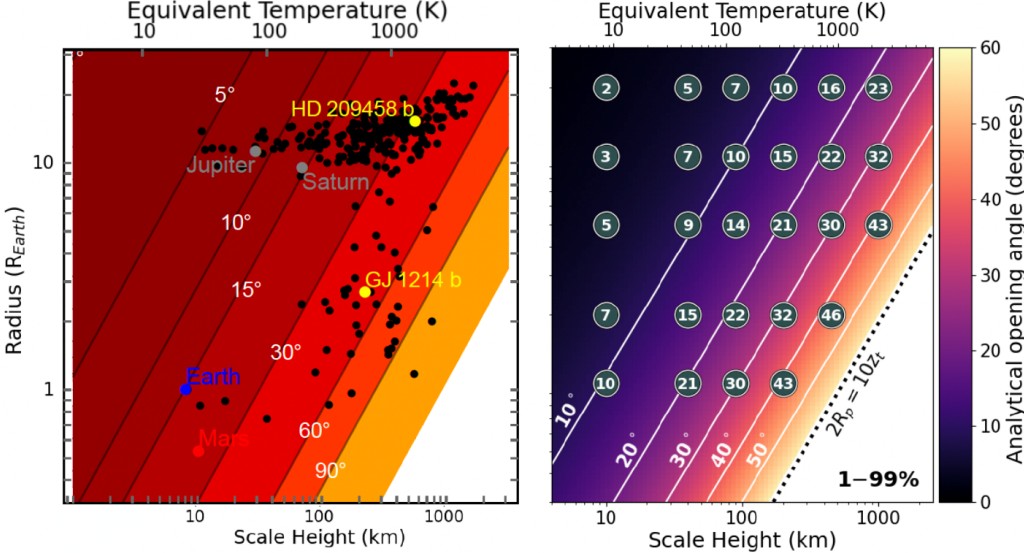

**Figure 7.** Opening angles according to the atmospheric scale height and the radius of the planet. (**Left**) Simple estimate of the opening angle of the region around the terminator that affects the transmission spectrum (i.e., the limb) [68]. The atmosphere is assumed to become transparent above the 1 Pa pressure level. Black dots are known planets for which the radius and surface gravity have been measured. (**Right**) Analytical calculation of the opening angle [17]. The values in the gray circles denote the opening angles (in degrees) that were geometrically calculated from the Monte-Carlo radiative transfer [17]. Despite the differences in these two plots, in particular for the large-scale height and radius, the trends are similar. For warm Neptunes, such as GJ1214b, or hot Jupiters, such as HD209458b, the opening angles are large (25° to 35°, respectively) and denote that the probe region extends significantly above the terminator of the planets.

This assumption has been used in the characterization of many exoplanetary atmospheres in low-resolution spectroscopy, including hot and ultra-hot Jupiters [8,47]. However, many recent studies place caveats on this assumption, especially for hot planets ($T_{eq} > 1000$ K) [13,16,18,68,69]. In transmission, we probe an angle around the terminator which can actually extend very much in the day and the night sides of the atmospheres [21]. Furthermore, we target in transmission the spectroscopy the upper part of the atmosphere at pressures of millibars to microbars, as the atmosphere becomes rapidly opaque at higher pressure. We show in Figure 7 the analytical opening angles which correspond to the side of the probe region as a function of scale height and planetary radii [17,68]. We see that for the hottest planets, this angle reaches high values (above 25 degrees), indicating that we are actually probing a region highly extended around the terminator. In addition, it has been demonstrate that depending on the atmospheric composition, this angle can increase even more for very hot atmospheres [14,16]. As we are probing the upper regions on the atmosphere in transmission, we are even more sensitive to the large scale height dichotomy in hot atmospheres, implying a larger opening angle. We note that this angle, which defines only a theoretical geometrical volume, can actually differs from the morning to the evening limb. Depending on the wavelength observed, we probe very different regions of the atmosphere due to thermal dissociation of species on the day side (e.g.,

water). Thus, both the night and the day sides of the atmosphere are probed making the transit spectrum highly impacted by the 3D structure of the atmosphere and its variability in term of thermal structure and composition. Furthermore, we know that the spectrum can significantly change during the transit, as is shown in Figure 8. This is a simulation of a transit of WASP-121b where we present the transmittance map and the spectra at different phases. The atmosphere was generated by a GCM [70], and the spectra were computed using Pytmopsh3R [18]. This simulation shows that the spectra can vary by hundreds of ppm between the ingress and the egress. These differences are due to the fact that WASP-121b's atmosphere is hardly asymmetric across and along the limb, with a day–night thermal difference, a large chemical dichotomy and an hotter evening limb compared to the morning limb due to jets.

Other processes may also play a role in the atmospheric composition and hence the opening angle, such as photochemistry. This chemical process is particularly important for closely orbiting planets receiving intense visible and ultraviolet flux from their star. Although photochemistry becomes negligible for planets with equilibrium temperatures above 1400 K [71], for warm to hot giant planets, it has an impact on atmospheric composition with strong contamination of the night side by species produced photochemically on the day side, implying compositional heterogeneities [71–74]. Photochemistry can also provide observational constraints. Tsai et al. [75] detected sulfur dioxide in the hot Jupiter WASP-39b, and they demonstrate that it has been photochemically produced as constrained by data from the JWST and informed by a suite of photochemical models. However, we must keep in mind that the data for the visible and ultraviolet absorption cross-section are still poorly known, especially at high temperatures, which are known to have a thermal dependence [76].

We summarize these effects in Figure 9, where we show how the orbital configuration may impact the spectra. Indeed, hot and ultra-hot Jupiters orbit very close to their stars and are most likely tidally locked, implying that the rotation of the planet during the transit is not negligible and can represent up to 30 degrees during the transit. We thus probe very different longitudes of the atmosphere, resulting in strong variations in the spectra. It is clear that considering a mean spectrum from the whole transit for hot Jupiters and ultra-hot Jupiters could lead to erroneous atmospheric characterization and that a temporal resolution during transit is needed to avoid such errors.

This problem can currently been solved by high-resolution spectroscopy. With this technique, we can resolve the absorption lines, implying that we can accurately calculate the Doppler shift in its lines to access to the radial velocity of the planets resulting from the orbit, the rotation rate and for more accurate instruments, even the speed of the exoplanet's winds [20,40]. We are thus able to plot radial velocity maps during the transit where each point corresponds to one spectrum. The temporal resolution is, however, limited by the signal-to-noise ratio, which is why we are only recently able to obtain information on the atmospheric winds, especially thanks to ESPRESSO, CRIRES+ [39,77] and, in the near future, with NIRPS spectrography [78]. A second advantage of resolving the lines in high-resolution spectroscopy is that it avoids degeneracy between species and thus allows for a better detection of molecular species compared to low-resolution spectroscopy observations where the bands can overlap. By combining these two advantages, we can spatially and temporally map each species detected in the atmosphere to gain a better understanding of the latitudinal and longitudinal distributions in its composition [40,42].

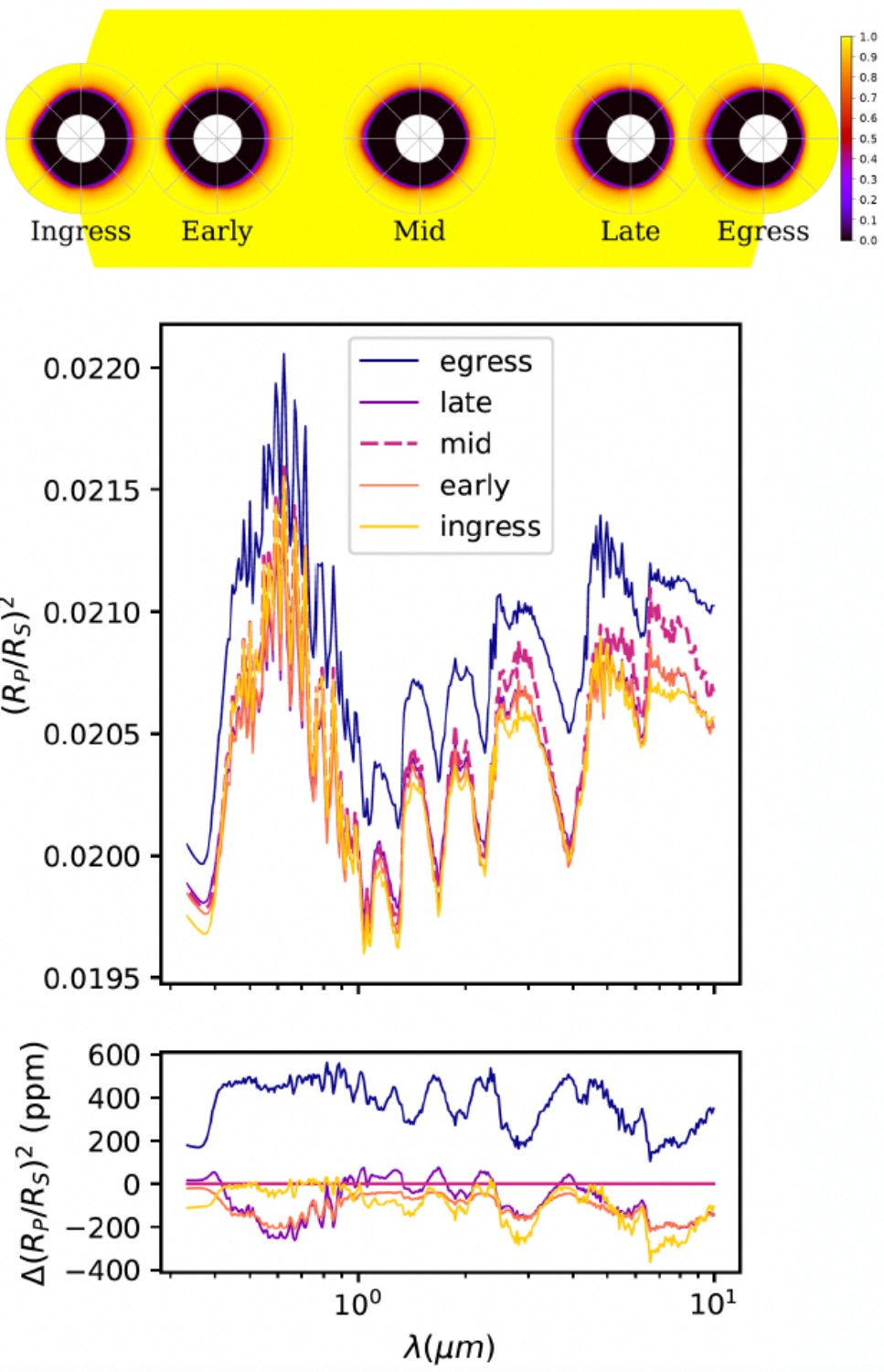

**Figure 8.** (**Top**) Transmittance maps of WASP-121b at 0.6 microns. The orbital phase angle is −15 degrees for ingress and +15 degrees for egress. For visual reasons, the planet's atmosphere has been enlarged with respect to its radius, and the early and late transmittance maps are slightly shifted. Only half of the planet covers the star at ingress and egress. (**Middle**) Spectral variations in the transit depth of WASP-121b during a transit for each phase shown above. (**Bottom**) The difference between each spectrum and the mid-transit spectrum, taken as a reference. During transit, the spectrum varies from tens to hundreds of ppm depending on the wavelength. Such differences are detectable by recent instruments, especially the JWST. Figure adapted from [18].

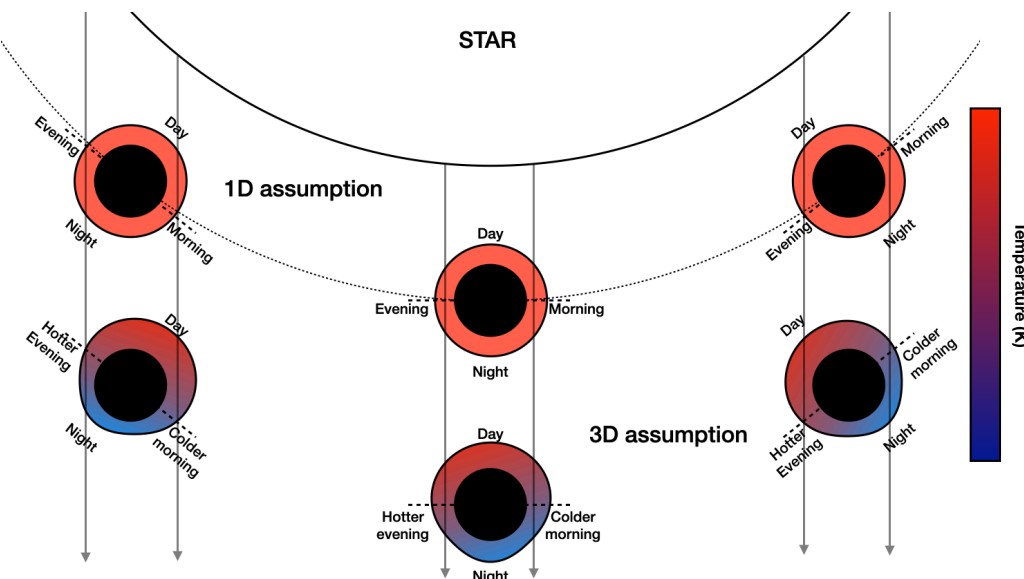

Varius regions probed considering 1D or 3D assumption for the atmosphere, with large temperature and scale height differences

**Figure 9.** Scheme of the transit at the ingress (**left**), the mid-transit (**middle**) and the egress (**right**) of an ultra-hot Jupiter considering a 1D (**top**) or a 3D (**bottom**) assumption to model its atmosphere. As the planet is tidally locked, we probe a colder region during the ingress than during the mid-transit due to the planet's rotation. We even probe the shifted hot spot at the egress due to this peculiar geometric configuration. It shows that during the transit, a significant part of the atmosphere is probed, extending significantly above the limbs, as suggested by the 1D assumption.

This method was used with ESPRESSO data on the ultra-hot Jupiter WASP-76b, showing that the iron varied temporally during the transit [21]. This result follows from the assumption of an asymmetric distribution of iron on the day side of the planet and its condensation at the limb, which created this blue-shifted radial velocity. Three-dimensional GCM simulations show that this trend in radial velocity could indeed be fitted by removing gaseous iron from the leading limb of their model and not changing the temperature structure [79]. However, similar results can be obtained by applying a large temperature difference between the trailing and the leading limbs of the planet [79]. Interpreting spectra that originate from inherently 3D exoplanetary atmospheres is thus a challenge, especially for hot and ultra-hot Jupiters. These extremely irradiated planets are characterized by large day–night temperature contrasts that drive (i) a complex atmospheric circulation pattern, (ii) an extreme contrast in the chemical composition (thermal dissociation, recombination, quenching, etc.) and (iii) extreme scale height variations, making them look like mushrooms [14,68,70]. Figure 9 shows this mushroom shape on hot and ultra-hot Jupiters predicted by GCM models [16,70,79] compared to the 1D assumption (e.g., a homogeneous atmosphere). This figure highlights both the intrinsic 3D structure of these atmospheres, in particular their strong day–night scale height dichotomy and the rotation rate during the transit due to the fact that these exoplanets orbit very close to their stars. This makes hot and ultra-hot Jupiters difficult to interpret with too simplistic models.

## 4. Updates on 3D Global Climate Models

Over the last few decades, the GCM has provided atmospheric models of many different exoplanets. There is a large variety of independent GCM models: LMDZ [80], ExoCAM [81], ROCKE-3D [82], THOR [83] or SPARC/MIT [67]. The first aim of these GCMs was to study atmospheres of our Solar System's planets (e.g., the Earth [81,82], Mars [80,84], Venus [85], Jupiter [86] and Saturn [3]). Many works have been conducted on cold giant planets thanks to Solar System exploration since the 1960s and more recently with the Cassini and JUNO missions. Local measurements of gravity, chemical composition, dynamics and pressure–temperature profile provide very accurate and global observational

constraints to improve the current GCMs. With the discovery of hot and ultra-hot Jupiters, GCMs had to be adapted to the particular conditions of these exoplanets. First, most of them are most likely tidally locked [87,88], which affects the energy balance and dynamics in many ways. Secondly, they are highly irradiated, which is very different from what we know within our Solar System, and they have different chemical, radiative and dynamic time scales in these atmospheres, which calls for new GCMs to be developed [67,83,89,90].

Theoretical work has demonstrated the importance of two fundamental lengths in atmospheric dynamics. First, the Rossby radius of deformation, which is the characteristic length scale at which the Coriolis force resists perturbations in pressure. It is defined as

$$R_d = \frac{NH}{f_c}, \tag{2}$$

where $N$ is the Brunt–Väisälä frequency governing the period of gravity waves, $H$ is the scale height of the atmosphere and

$$f_c = 2\Omega \sin \phi \tag{3}$$

is the Coriolis parameter governing inertial motions, with $\Omega$ being the angular rotation rate of the planet and $\phi$ the latitude [91]. Second, the Rhines length, defined as

$$(U/\beta)^{1/2}, \tag{4}$$

is the scale at which the planetary rotation causes strong east–west winds, also called jets. $U$ is the characteristic speed of the horizontal wind and

$$\beta = 2\Omega \cos(\phi/a) \tag{5}$$

is the longitudinal gradient of the planetary rotation, where $\Omega$ and $\phi$ are the same as defined above and a is the radius of the planet. We know that turbulence tends to be horizontally isotropic at small scales, but at large scales, when we reach the Rhines length, the turbulence tends to evolve preferentially in the east–west direction, implying the development of strong zonal jets [88]. Thus, at the large-scale heights and moderate rotation rates of hot Jupiters, the Rossby radius of deformation and Rhines length are typically of the order of the planetary radius of the planet, implying planet-wide dynamics, unlike the giant planets of the Solar System.

To illustrate the impact of these particular atmospheric behaviors on hot Jupiters, we show in Figure 10 a set of eight GCM simulations of equilibrium temperature from 1400 K to 2100 K [16,70]. On the right side of the figure, we show the latitude–longitude temperature map at 1 mbar compared to the equatorial cut temperature map plotted in altitude (km). The left panels highlight the strong day–night asymmetry and reveal the "mushroom" shape of the atmosphere, which increases with the increasing equilibrium temperature. They also show the hot spots in these atmospheres, which are shifted eastward by ~30 degrees to ~10 degrees from $T_{eq}$ = 1400 K to $T_{eq}$ = 2100 K, respectively. Indeed, the hotter the planet, the smaller the radiative time scale compared to the dynamic time scale, resulting in a smaller hot spot shift despite the super fast zonal jet (a few km/s). The right panels highlights the asymmetry along the limb, where we clearly see the hotter morning and the colder evening (as illustrated in Figure 9).

These models, however, remain incomplete, as described in [14]. For instance, they assume the thermal dissociation of species which occurs in such hot atmospheres, but they do not take into account the recombination of $H_2$ on the night side, which tends to homogenize the day–night temperature transition [92]. In addition, the chemical equilibrium is assumed when we know that kinetics or out-of-equilibrium chemistry could occur in such atmospheres with a non-negligible impact on the radiative transfer and on the emission or transmission spectra [15,76]. Furthermore, another caveat concerns the spectroscopic data used in the models, either 1D or 3D. Several databases contain data for many atoms,

molecules and ions such as HITRAN [93,94] or EXOMOL [95–97]. However, these databases are limited in the range of temperature and pressure because it is hard to replicate the conditions of hot Jupiters and ultra-hot Jupiters in the laboratory. We thus use extrapolated spectroscopic data, which could lead to biases or erroneous interpretations, in particular in terms of abundances. It can also have a dramatic impact on the energy balance calculated by the radiative transfer in the atmospheric model (1D or GCMs). Some caveats on the impact of the databases to fit the observations also exist [98,99].

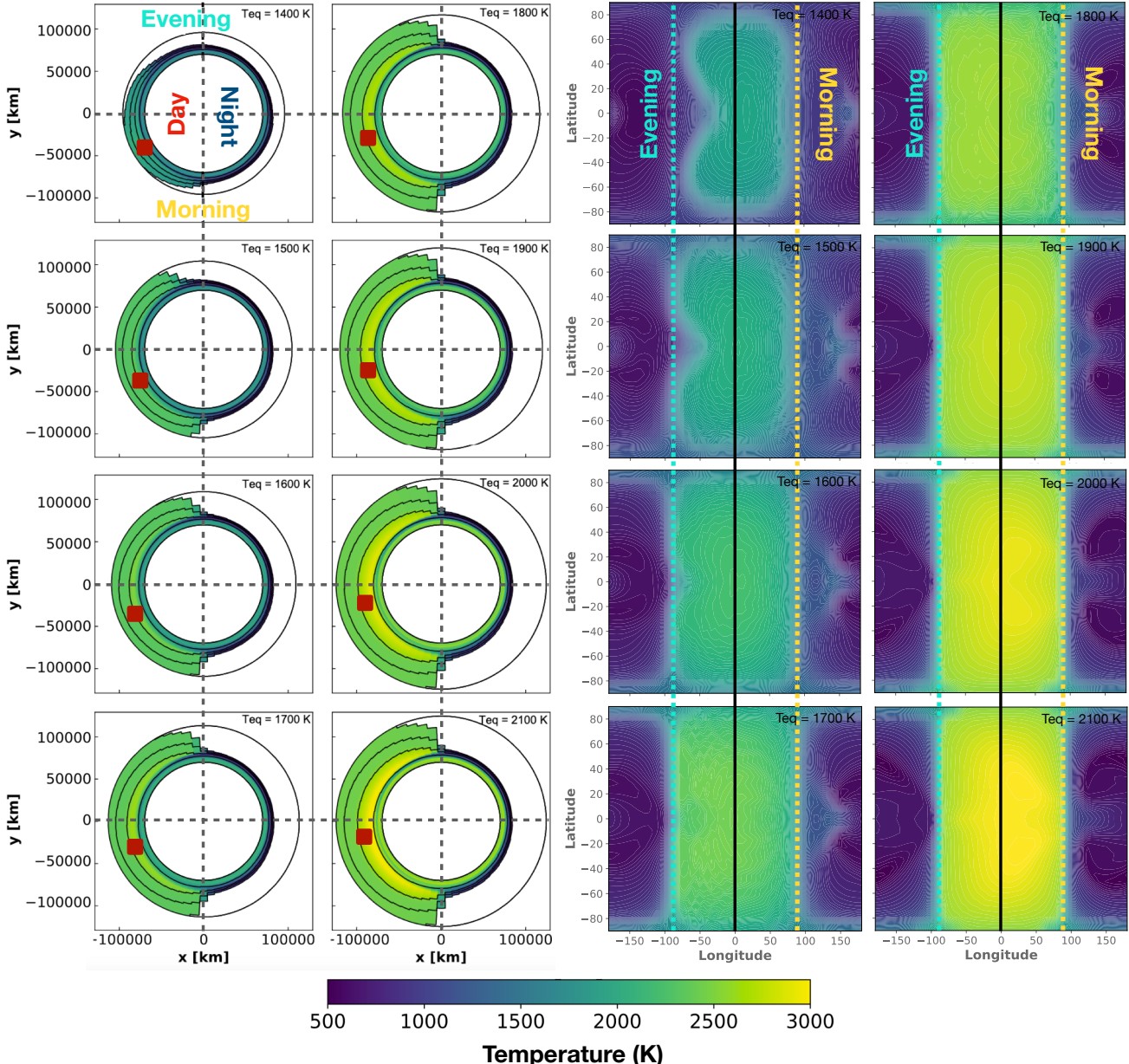

**Figure 10.** (**Left panels**) Equatorial cut of the temperature for 8 GCM simulations. From the center outward, the five solid black lines are the $1.434 \times 10^7$, $10^3$, 1, $10^{-2}$, and $10^{-4}$ Pa pressure levels. The hotter the equilibrium temperature, the larger the day–night thermal and chemical dichotomy. The red points show the hot spots of each simulation. (**Right panels**) Temperature map at 100 Pa of the corresponding GCM simulations. The black line shows the sub-stellar longitude. The blue and yellow dashed lines show the evening and the morning limbs, respectively. It thus highlights the hot spot shift and the limbs' temperature asymmetries. The equilibrium temperature of the planet (ranging from 1400 to 2100 K) is specified. Figure adapted from Pluriel et al. [16].

To overcome these uncertainties, observational constraint are needed. Firstly, we can access the wind speed almost all around the planet thanks to the transit observations in high-resolution spectroscopy with accurate enough sprectrograph, such as Spirou, ESPRESSO, CRIRES+ and NIRPS [100,101]. Indeed, GCM models remain unconstrained, for instance, on the drag, which requires more observations of hot Jupiters and ultra-hot Jupiters to be fully modeled [102]. Having more accurate phase curves would also help in obtaining more accurate information on the redistributed energy in the atmospheres and could also constrain our knowledge on the cloud coverage of the planet, which has a major impact on its atmospheres. The detection of species with accurate abundances would also help to determine which regime the atmosphere is in (e.g., equilibrium, out of equilibrium, with quenching, etc.), which has an important impact on the GCMs, as shown in [11,54]. We know for instance that optical absorbers (such as TiO, VO, Fe, Mg or ionic hydrogen) will create a strong thermal inversion in the stratosphere, resulting in large differences in the simulated atmospheres [16,103]. We have learned from observation that the temporal flux variability of hot Jupiters and ultra-hot Jupiters remains weak, around 2% maximum, indicating a great temporal stability, which was not always predicted by GCMs [104]. A final caveat concerns the models themselves. As explained in this section, the radiative and dynamic time scales are very different for hot and ultra-hot Jupiters compared to colder planets, which impact the time convergence of GCMs. Even if the models demonstrate a good agreement between each other, in particular on the hot spot shift due to one large eastward jet stream [50,105–107], a few studies have demonstrated that performing long convergence GCM simulations for hot Jupiters tends to converge on atmospheres with two jet streams at each tropic instead of one large equatorial jet stream [104,108]. This is typically the situation where observational constraints, especially on the winds, could bring substantial information to confirm or refute these simulations.

## 5. Interpretation of the Data: Retrieval Analysis

The interpretation of the data (phase curve, emission or transmission spectra) is a tricky yet crucial part of atmospheric characterization. Indeed, we need to know very well the models which are used to perform this interpretation and their limits to avoid misinterpretation. There are many methods to interpret the data, such as comparison with GCM simulations, as developed in Section 4, but we will focus in this section on Bayesian retrieval analysis.

Bayesian retrieval analysis is a massively used method to interpret phase curve, emission and transmission spectra thanks to retrieval codes such as Nemesis [109], petit-CODE [110], ARCIS [111] or TauREx [112,113]. Bayesian retrievals present two main limits: they need to compute millions of models to converge to stable posterior distributions, which requires massive computational power and time, and they need to exclude many parameters to avoid degeneracies. This is why almost every retrieval code developed so far was using 1D forward models; otherwise, they would require a long time to converge to a stable solution.

As we saw in Sections 3 and 4, exoplanetary atmospheres are 3D, and we now have space- and ground-based instruments accurate enough to possibly obtain elementary abundances, metallicity, wind maps, pressure–temperature profiles, etc. To be able to use the full potential of these new instruments, we need to adapt our interpretation models. It has been demonstrated that 1D assumptions in retrievals lead to overestimation by several orders of magnitude in the abundances, in particular in terms of the C/O ratio [14]. In transmission spectroscopy, the features on the spectrum for JWST observations comes from very different regions which have large scale heights and compositional differences that cannot be handled by 1D retrieval models. The analyses of a wide range of exoplanets from hot Jupiters to ultra-hot Jupiters has demonstrated that 1D retrieval models bias the retrieved abundances for planets with equilibrium temperatures above 1400 K [16]. The strong day–night dichotomy in the thermal structure and chemical composition of exoplanetary atmospheres are responsible for this bias. What is even more worrying is that these models produce

very good fits, but with incorrect parameters in the posterior distribution (in particular, the abundances and the temperature profiles). Therefore, simulation-based studies are needed to calibrate the optimal usage regime for a particular type of model [16].

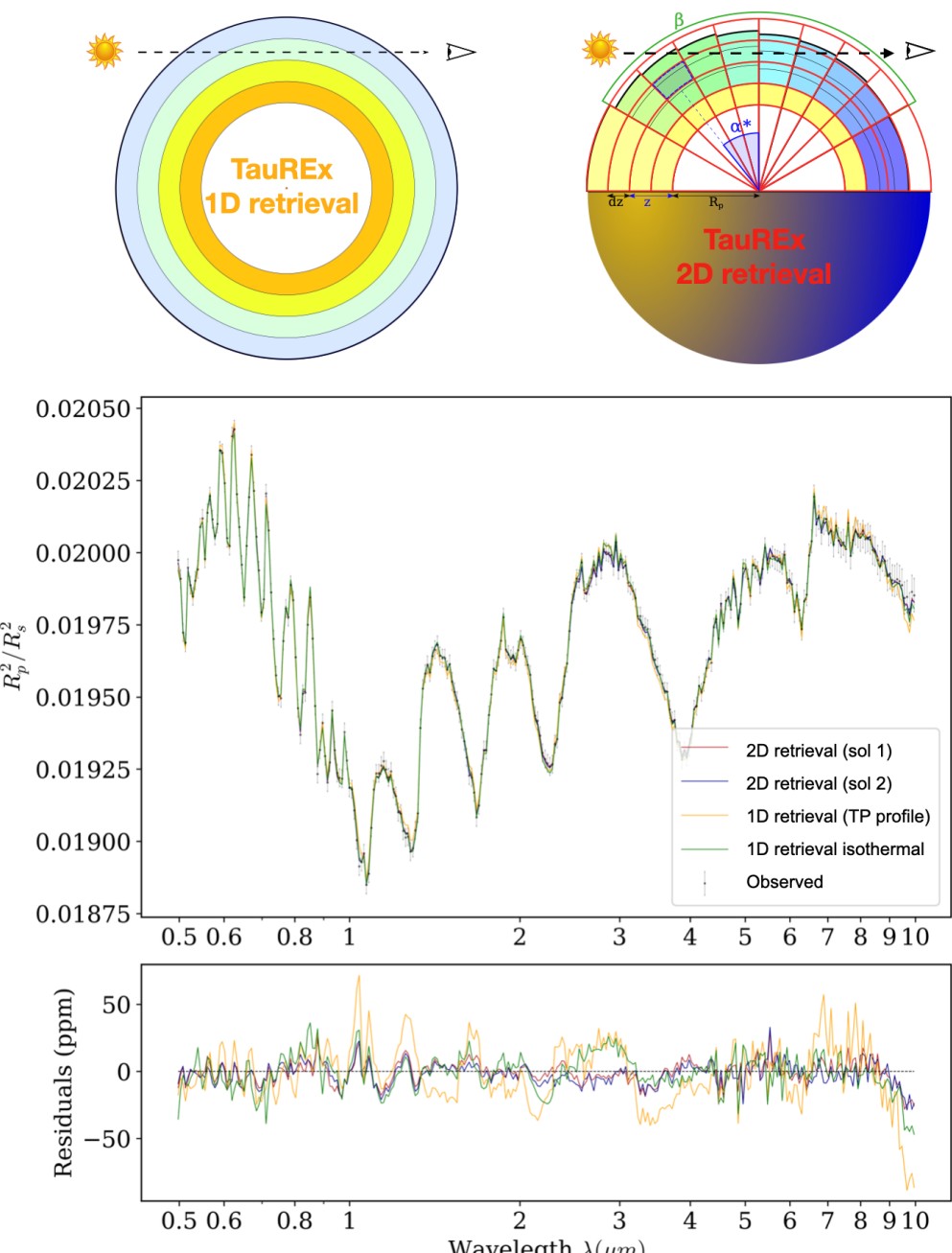

**Figure 11.** (**Top**) Visual representation of the 1D model (left) only relies on altitude and 2D model (right) using a polar grid whose the radial axis is the altitude and whose angular axis follows the solar altitude angle ($\alpha*$). (**Middle**) Retrieved transmission spectra of simulated WASP-121b (black dots) from the 1D model using isothermal TP profile (green) and 4-point TP profile (yellow). The two solutions from the 2D model are shown in blue and red. (**Bottom**) Residuals between the observed spectrum and the retrieved spectrum in ppm. Figure adapted from [18,114].

It has also been shown that the east–west asymmetry in hot Jupiters may have a strong impact on the retrievals [13]. This could imply over-estimations of the terminator temperatures given in the literature, which could also affect the chemical compositions inferred from these temperatures. Indeed, in many atmospheres, we are close to thermo-

dynamical limits, and a misinterpretation of the temperature could modify the dominant species or allow for quenching. The impact of the chemistry has also been analyzed [15], demonstrating that the kinetics have an important impact on the transmission spectrum. Planets with effective temperatures below 1400 K tend to have horizontally homogeneous, vertically quenched chemical compositions, while hotter planets exhibit large compositional day–night differences for molecules such as $CH_4$, for instance. Thus, retrieval models should take this effect into account to avoid retrieving incorrect abundances.

To avoid these biases or erroneous parameters, several new models have been developed in recent years from multi-1D to fully 3D models. A fully 3D model computes the atmosphere following a spherical coordinate system (altitude, latitude, longitude). A 2D model uses a polar grid whose radial axis is the altitude and assuming latitudinal or longitudinal symmetry. Finally, we call multi-1D (sometimes called 1.5D) models those which fill a latitudinal or longitudinal grid running several temporal 1D models. We illustrate in Figure 11 how the two-dimensional retrieval model is sufficient to unravel biases for ultra-hot Jupiters. In this figure, we compare retrieval analyses on a simulated JWST transit spectrum of WASP-121b [14] based on a GCM simulation [70] using the 1D version of TauREx [113,115] and the newly developed 2D parametrization of TauREx across the limb [114]. The simulated atmosphere is composed of $H_2$, He, $H_2O$, CO, TiO and VO, with solar abundance. The top panel shows the visual representations of the 1D and the 2D models (equatorial cut). For the 2D model, the temperature is defined in a 2D coordinate system $(\alpha^*, P)$ of which one dimension is angular and the second is pressure-based. This coordinate system is shown by the underlying (black) grid in Figure 11. The temperature, defined by Equation (6), follows a linear transition between $-\beta/2$ and $\beta/2$ [17,68]:

$$
\begin{cases}
P > P_{iso}, T = T_{deep}, \\
P < P_{iso}, \begin{cases}
2\alpha^* \geq \beta, T = T_{day}, \\
2\alpha^* \leq -\beta, T = T_{night}, \\
-\beta \leq 2\alpha^* \leq \beta, T = T_{night} + (T_{day} - T_{night})\frac{\alpha^* + \frac{\beta}{2}}{\beta}.
\end{cases}
\end{cases}
\tag{6}
$$

This equation relies on three temperature variables ($T_{day}$, $T_{night}$, $T_{deep}$), an angle parameter ($\beta$) and a pressure level defining the upper limit of an isothermal annulus ($P_{iso}$). The bottom panel of Figure 11 shows the retrieved spectra with the two models and the residuals compared to the simulated transit. Interestingly, every fit is very good and remains below 60 ppm in error, which is around the best level of noise expected for WASP-121b. This indicates that for an observer analyzing real data, these solutions are degenerated; thus, we cannot decide which model should be selected. It has been shown that a 2D parametrization managed to unravel the large C/O overestimation in hot Jupiters and ultra-hot Jupiters [114] thanks to a simple parametrization across the limb, whereas the 1D models failed by several order of magnitudes. We note that the two solutions found are due a degeneracy between the $\beta$ angle and the day-side temperature. For the model, it is equivalent to have a sharp transition between the day and night (small $\beta$) with a colder day side than to have a smother transition (larger $\beta$) with a hotter day side.

Many improvements to the TauREx code have been implemented in recent years, enabling it totake into account two chemical layers [12], to conduct phase curve retrieval [65] or to take into account equilibrium chemistry [116]. We highlight the example of TauREx-2D to demonstrate that the interpretation of data with retrieval analysis is still challenging, especially due to the high level of degeneracy that occurs in the models. Most of all, it is possible to find very good agreement between a retrieval model and the data, but with very incorrect parameters, either in abundances [14,16] or in terms of the thermal structure [114].

The community is aware of these caveats and continues to develop their tools to be able to analyze the coming data from JWST, NIRPS and future, more accurate instruments, especially having in mind the importance of the 3D structure of hot exo-atmospheres. Several teams are working on fully 3D parametrization techniques, such as TRIDENT [117], which is particularly insistent on the important differences observed in the transmission

spectrum due to the cloud coverage with or without assuming a 3D parametrization. The AURA code [118] also uses a 3D parametrization which is actually close to TauREx-2D but is also generalized in latitude. Thanks to these more complex codes, it is possible to find abundances and a thermal structure that are consistent with the GCMs. We present in Figure 12 a summary of the different geometries, as a function of equilibrium temperature and the presence or absence of an optical absorber, required in retrieval codes to limit the biases, errors and misinterpretations. This summary is an update on Pluriel et al. [16], with the most recent retrieval code improvement explained being in the previous paragraph. It is clear that for hot Jupiters and ultra-hot Jupiters that a more complex geometry should be assumed to perform consistent retrieval analysis. However, we need to be careful to use the adapted geometry depending on the planet configuration, equilibrium temperature and composition to avoid misinterpretation. Indeed, in the case of planets where the 3D impacts are not supposed to be significant (low opening angle, no optical absorber detected or low equilibrium temperature), it is cautious to first perform 1D retrieval and eventually to compare it with 3D retrieval. From two similar fits, the simplest model should always be privileged (Occam's razor principle) to avoid misinterpretation. In addition, it is not efficient to perform 3D analysis when assuming a 1D one will give similar results.

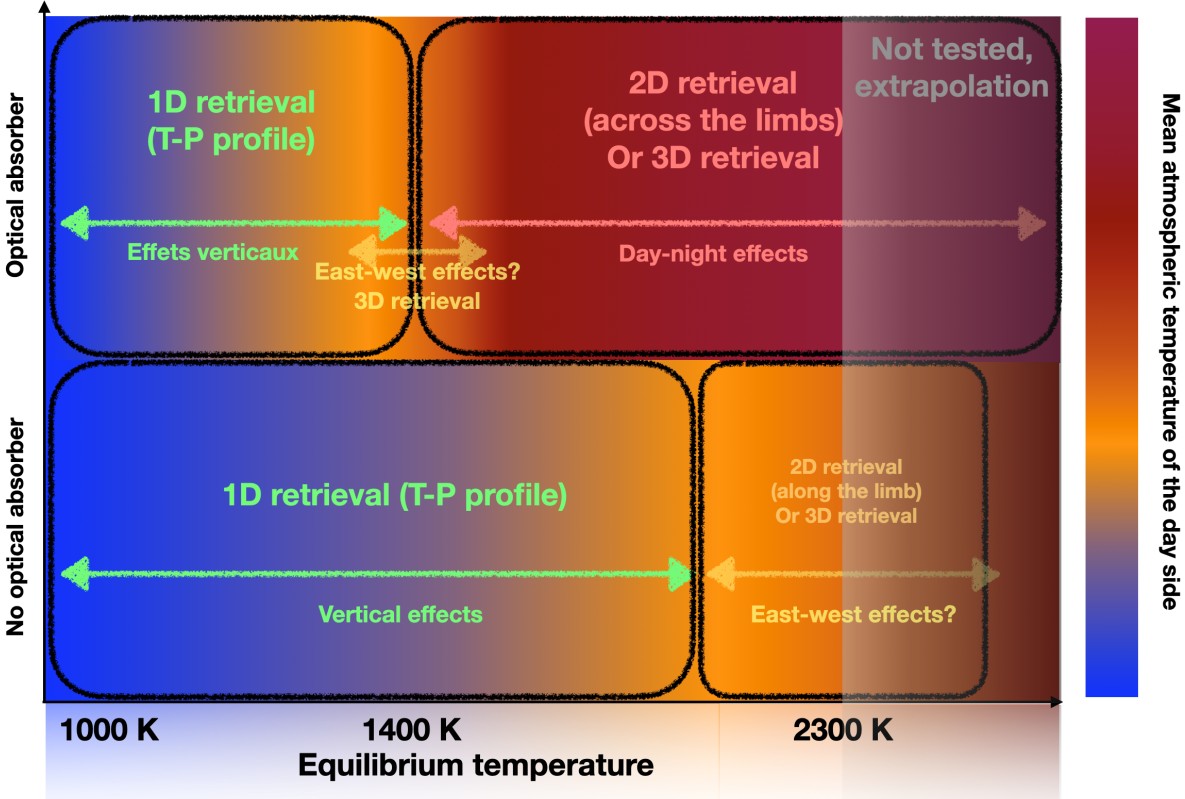

**Figure 12.** Summary of the different geometries required in retrieval codes to avoid biases as a function of the equilibrium temperature of the planet and the presence or absence of optical absorbers (hence, thermal inversion). One-dimensional retrieval models appear to provide relatively satisfactory parameter estimates for planets with equilibrium temperatures lower than 1400 K when optical absorbers (TiO, VO, K, Na, metals, ionized hydrogen, etc.) are present in the atmosphere. However, they lead to biased parameter estimates above this limit, where 2D or 3D retrieval codes are mandatory. When no optical absorbers are present in the atmosphere, the validity of the 1D retrieval code extends to an equilibrium temperature of 2000 K. Above this temperature, the estimated parameters become biased, probably due to east–west effects, which require 2D or 3D models to unravel the complexity of the spectrum. However, we suggest clues for the effect of the east–west asymmetry, and further investigations are needed to quantify their effects. Figure adapted from Pluriel et al. [16].

Finally, a similar caveat discussed in Section 4 on the spectroscopic data also has to be taken into account for retrieval analysis. As we explained it, the spectroscopic databases lack consistent data for the hottest atmospheres, which implies that the cross-sectional data may overestimate some features or underestimate others. As every model intrinsically assumes that their spectroscopic data correspond exactly to the real spectroscopic features, differences between the data and the ground truth could have a non-negligible impact on the retrieved abundances, cloud coverage or thermal structure of the atmosphere.

## 6. Ways Forwards

Hot exoplanetary atmospheres represent a great opportunity for atmospheric characterization. These targets are easier to observe due to their large atmospheres compared to cold exoplanets, and they have a higher probability to transit in front of their star, which increases the number of methods applicable to analyzing their atmospheres. Hot and ultra-hot Jupiters represent a laboratory where we can test atmospheric theories developed on their well-studied, cooler counterparts. They can answer questions such as the following: How does atmospheric chemistry respond to intense thermal heating? How and when do clouds form in atmospheres? What processes control a climate's circulation? Answering these questions in hot and ultra-hot Jupiters will open the door to understanding the nature of exoplanets as a whole, and all of these questions drive us towards a full mapping of their atmospheres, from their day sides through their night sides. These planets are greatly non-homogeneous, with very large chemical and thermal contrasts across their atmospheres. In this context, to unfold the complexity of hot and ultra-hot Jupiters' atmospheric behaviors and their evolution, it is mandatory to take into account the 3D effects.

The characterization of hot exoplanetary atmospheres also remains a challenge. The new instruments recently installed will bring many improvements over what has been achieved so far. In high-resolution spectroscopy, the precision of the radial velocities will reach the cm/s mark, and thanks to the VLT, the signal-to-noise ratio has become very high, even for low-magnitude targets. The temporal resolution of the phase curve and the transit (when present) will then allow for the characterization of a large part of the atmosphere, as summarized in Figure 9. High-resolution spectroscopy instruments can also currently observe infrared wavelengths, allowing for new absorption lines to be reached, especially molecular ones, thus obtaining new detections. In low-resolution spectroscopy, the JWST and its 6.5 m mirror will also allow for a significant improvement in signal-to-noise ratio, making it possible to accurately determine the elemental abundances of chemical species. In addition, the very wide wavelength coverage in the infrared spectrum will not only allow the discovery of new species, but will also break the degeneracies between species.

Despite these impressive observational improvements, we have also seen that the intrinsic 3D structure of hot exoplanets makes it difficult to interpret the observations because of model assumptions. Firstly, retrieval models using the Bayesian method mostly use simplistic 1D assumptions in order to be run in a reasonable time, but this results in errors and biases in the results provided. New models using less simplistic assumptions (2D to 3D) have emerged and seem to be able to disentangle some of the biases observed with 1D models. That said, the parameterizations used remain simple, as it is not yet possible to use GCMs as forward models to perform Bayesian analysis. Secondly, we have seen that GCMs describing hot and ultra-hot Jupiters are not sufficiently constrained yet. Indeed, unlike retrieval analysis, GCMs contain all the physics, chemistry and dynamics needed to describe an atmosphere (they also use approximations and consider every process, but they are way more complete than models used in retrieval codes). However, the high level of detail provided by these models is hardly constrained by observations and results in many degeneracies. For instance, we have no observational constraint of the interior of these planets despite knowing that it has a major impact on the upper atmosphere, as demonstrated in the giant planets of our Solar System. We have seen that a number of observational constraints (wind measurements, atmospheric composition,

species abundance, cloud cover, albedo) allow us to distinguish between models and better understand the physics, chemistry and dynamics of these planets. A better understanding of the giant planets of our Solar System could therefore be very useful for improving our knowledge of the atmospheres of exoplanets. The level of accuracy in all the in situ measurements of the TP profile (as performed by Cassini on Saturn), of the love number (JUNO mission) and of the dynamics, clouds coverage, etc., cannot be reached in exoplanets; this limitation thus encourages comparison among planets.

The study of the structure and circulation of the atmosphere is therefore a challenge. We have observed so far hot and ultra-hot Jupiters, irradiated brown dwarfs, Young Giants (observed in direct imaging) and giant planets in out Solar System. These four categories covers extreme ranges of key parameters, from very low to very high irradiation and interior heat flux. Interestingly, there is continuity in these key parameters, which presents great synergy between these categories. Therefore, the information obtained in some categories also improves the knowledge of the other categories [119,120].

**Funding:** This research was funded by the National Centre for Competence in Research Planet supported by the Swiss National Science Foundation (SNSF), project 200021_200726.

**Institutional Review Board Statement:** Not applicable.

**Informed Consent Statement:** Not applicable.

**Data Availability Statement:** Not applicable.

**Acknowledgments:** This project has been carried out in the frame of the National Centre for Competence in Research PlanetS supported by the Swiss National Science Foundation (SNSF). WP acknowledges financial support from the SNSF for project 200021_200726. I would also like to warmly thank David Ehrenreich and Tiziano Zingales for their very useful comments on this review.

**Conflicts of Interest:** The authors declare no conflict of interest.

## Abbreviations

The following abbreviations are used in this manuscript:

| | |
|---|---|
| GCM | Global climate model |
| JWST | James Webb space telescope |
| HST | Hubble space telescope |
| IRAC | Infrared array camera |
| VLT | Very large telescope |

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
