# Peer review of "Hot Exoplanetary Atmospheres in 3D"

_remotesensing, doi:10.3390/rs15030635_

Round 1
Reviewer 1 Report
This is an excellent review which I have read with interest and pleasure. It gives you in a concise and simple form the state of the art and the outstanding problems. While it won't be particularly stimulating for specialists, for scientists in close fields it might be very useful.
The paper is a rare case when it doesn't require any revision, even the few minor spell errors can be overlooked.
Author Response
I thank the referee for is response. My aim was indeed to target this journal at scientists in related but non-expert fields, I am glad that it is received as such.
As no scientific changes were required, the manuscript remain very similar. I just proofread the English.
Reviewer 2 Report
None at this time. See comments for the Editors.
Author Response
I thank the referee for this concise report. I proofread English as required.

Reviewer 3 Report
The present content of the paper seems very good, the English is clear for readers, for which it is not native language like me. The situation concerning the thematic is very clear described in the course of the present used instruments and observations as well as in the course of the theoretical models and their application to observations. There is also a good discussion regarding the using of new instruments and particular these on the board of the James Webb space telescope.
A small (and not strong_ recommendation
I think that a short discussion about the possible effects caused by photochemistry processes over the observed gaseous compounds could be interesting and useful. The hot and ultra hot giant planets are usually too close orbiting around the corresponding stars and the effects from stellar X and UV radiation as well as their time variations (cyclic or not) could be significant for the models improvements and (may be) detectable by observations.
Author Response
I thank the referee for its report.
The remark on photochemistry process is very relevant. That's indeed important to take into account photochemistry process for such hot atmospheres. I thus added a short discussion in section 3.2 to explain the impacts of this process on the models and on the observables.
Even if the English was good enough to referee, I proofread it as required by an other referee.
Round 2
Reviewer 2 Report
Dear Editor,
the revisions of the manuscript addressed initial concerns about the presentation. The review is broad and comprehensive, and it represents a useful addition to available literature on the topic.